# Immunotherapy for Elderly Patients with Advanced Non-Small Cell Lung Cancer: Challenges and Perspectives

**DOI:** 10.3390/ijms26052120

**Published:** 2025-02-27

**Authors:** Anass Baladi, Hassan Abdelilah Tafenzi, Othmane Zouiten, Leila Afani, Ismail Essaadi, Mohammed El Fadli, Rhizlane Belbaraka

**Affiliations:** 1Department of Medical Oncology, Mohammed VI University Hospital, Marrakech 40000, Morocco; hassanabdelilah.tafenzi@gmail.com (H.A.T.); drzouitenothmane@gmail.com (O.Z.); afanileila@gmail.com (L.A.); elfadli.mohamed2000@gmail.com (M.E.F.); belbaraka.r@gmail.com (R.B.); 2Laboratory of Biosciences and Health, Faculty of Medicine and Pharmacy, Cadi Ayyad University, Marrakech 40000, Morocco; ismail_onco@yahoo.fr; 3Medical Oncology Department, Avicenna Military Hospital, Marrakech 40000, Morocco

**Keywords:** lung cancer, elderly patients, immune checkpoint inhibitors, non-small cell lung cancer, immunosenescence

## Abstract

Lung cancer, a leading cause of cancer-related mortality, disproportionately affects the elderly, who face unique challenges due to comorbidities and reduced organ function. Immune checkpoint inhibitors (ICIs) offer a more tolerable alternative to chemotherapy, but their efficacy and safety in elderly non-small cell lung cancer (NSCLC) patients remain underexplored due to limited representation in clinical trials. A narrative literature review was conducted using PubMed, Embase, and the Cochrane Library to evaluate studies on elderly NSCLC patients (≥65 years) treated with ICIs. Key outcomes assessed included overall survival, progression-free survival, response rates, treatment-related adverse events, and the influence of immunosenescence on treatment efficacy. The review highlighted evidence supporting the efficacy and safety of ICIs in elderly NSCLC patients, particularly those with good performance status. Age-related immunosenescence may affect outcomes, emphasizing the need for individualized treatment. Limited data suggest ICIs alone may be preferable to chemo-immunotherapy for patients over 75 years. However, the exclusion of elderly patients from clinical trials and methodological limitations reduces the generalizability of these findings. ICIs hold promise for advanced NSCLC in older adults, but tailored approaches and greater elderly inclusion in trials are needed to optimize outcomes.

## 1. Introduction

Lung cancer is a significant health concern worldwide. In 2020 alone, nearly 1.8 million deaths were reported out of 2.2 million newly diagnosed cases [1,2]. The age-standardized incidence rate of lung cancer, which measures the number of new cases per 100,000 individuals each year, shows considerable variation across different countries. For instance, Denmark reports a rate of 36.8 per 100,000, while Mexico has a considerably lower rate of 5.9 per 100,000 [3,4]. This disease is a significant concern in developing countries like China, where rates have increased in people between 50 and 59 years [1]. Lung cancer predominantly impacts older individuals, with the median age at diagnosis being approximately 70 years [5]. Over the past two decades, the incidence of lung cancer has shown a consistent increase with advancing age [6]. Research indicates that the frequency of lung cancer cases differs depending on the geographical location and is impacted by variables like tobacco use [7]. In addition, older lung cancer patients have a higher frequency of comorbidities and frailty indicators. These indicators include performance status, independence in activities of daily living, and mobility. The genomics of lung cancer are being developed, making it possible to adopt targeted and personalized treatments for each subgroup of patients. Among the means of reducing mortality, screening programs in high-risk groups have proved their worth. But it remains true that prevention is the most effective approach to reducing the incidence of lung cancer, through smoking cessation programs and public education [8,9,10,11].

In advanced lung cancer, immune responses to immunotherapy vary with age. Older patients (50+ years) generally have better tumor immune environments than younger patients [12]. As a result, older patients often have a stronger response to immune checkpoint blockade therapy, leading to longer survival without disease progression [13]. Studies indicate that ICIs, such as nivolumab, provide comparable overall survival (OS) benefits to younger populations, with a median OS of 23 months reported in elderly patients receiving nivolumab [14]. Furthermore, combining ICIs with platinum-based chemotherapy has improved both OS and progression-free survival (PFS) in treatment-naïve older adults, with a hazard ratio of 0.78 for OS [15]. The use of ICIs in older patients with non-small cell lung cancer (NSCLC) has shown both effectiveness and safety [16]. However, due to the toxicities of chemotherapy components, questions have been raised about the safety of chemo + ICI-based combos in the elderly [17]. Overall, as immunotherapy can help the elderly population, age should not be viewed as a barrier to receiving ICI treatments [18]. Indeed, pivotal clinical trials have often underrepresented older patient groups, raising concerns about the applicability of the results to elderly populations. To address this gap, the academic ELDERLY trial was initiated in France. This study was designed to carefully evaluate the efficacy and safety of atezolizumab in elderly patients with advanced non-small cell lung cancer. Participants received chemotherapy with monthly carboplatin and weekly paclitaxel, representing a significant advancement in the treatment of this often-overlooked population [19]. Immunotherapy has become an important approach in treating lung cancer, particularly advanced NSCLC. While traditional chemotherapy and targeted therapies have limited long-term effectiveness, checkpoint inhibitors like PD-1 (programmed cell death protein 1), a receptor on T cells that regulates immune responses, and PD-L1 (programmed death-ligand 1), a protein expressed on tumor cells that helps them evade immune detection, have shown promise by improving response and survival rates. Lung cancer’s high immunogenicity, especially in cases with P53 gene mutations, makes it more responsive to immunotherapy. Combining checkpoint inhibitors with radiation or other immunotherapies is a strategy being explored to overcome resistance. Overall, immunotherapy has revolutionized lung cancer treatment and remains a key area of ongoing research [20,21,22,23,24]. The available evidence on immunotherapy’s effectiveness in older lung cancer patients is limited, as they are often excluded from clinical trials due to health conditions, frailty, and lower overall function. Some research suggests that older patients experience similar benefits from immunotherapy as younger patients and tolerate it well [25]. It is crucial to include older adults in clinical trials to understand better the benefits and risks of immunotherapy in this population [26].

Cancer and aging are closely related processes, both at the molecular and epidemiological levels. Research has shown that aging and cancer share common epigenetic replication patterns, such as the age-related increase in the Cell DRIFT signature, a specific epigenetic marker characterized by age-related DNA methylation changes that distinguish cancerous tissues from normal tissues. This signature is particularly relevant in elderly patients, as it may help explain age-associated tumor behavior and predict responses to therapies such as immune checkpoint inhibitors (ICIs) [27]. Age-related changes in the tumor microenvironment, such as the accumulation of senescent fibroblasts, promote the growth of tumors. The senescence-associated secretory phenotype (SASP) is a cellular state associated with senescent cells that secrete a mix of pro-inflammatory cytokines, chemokines, growth factors, and proteases. SASP plays a dual role in cancer progression, promoting tumor growth and metastasis under certain conditions, while also influencing the immune response to therapy [28]. Age-related malignancies also arise and grow due to somatic mutations, age-related changes in gene expression, and mechanisms for repairing DNA damage [29]. In general, enhancing treatment approaches for age-related cancers and creating focused therapies depend heavily on our ability to comprehend the molecular connections between aging and cancer.

Cancer management in the elderly is complicated by the heterogeneity of the elderly population, physiological differences from younger patients, and the lack of data from randomized controlled trials [30]. Aging has an impact on various aspects of cancer, including incidence, progression, prognosis, therapeutic options, and psychosocial aspects [31]. Age-related changes in cells, tissues, immune function, and general physical condition influence cancer development and pathology [32]. Understanding the relationship between aging and cancer is critical due to the predicted increase in cancer incidence in older populations. Patients with lung cancer who are older are more likely to have comorbidities and signs of frailty, especially when it comes to functioning in daily life, autonomy in their activities, and mobility. It is not yet well-defined whether the therapeutic strategies for older adults diagnosed with lung cancer are clear or not, as a lot of studies have disregarded this particular age group [33]. A comprehensive geriatric assessment plays a major role in developing an appropriate approach for the older adult patient, which will lead to a decrease in unwanted adverse effects and treatment failure. Consequently, early identification of active frailty evaluation and intervention will result in many therapeutic options available for elderly patients with lung cancer.

## 2. Materials and Methods

A narrative literature review was carried out to examine the application of immune checkpoint inhibitors (ICIs) in older adults (aged 65 and above) with non-small cell lung cancer (NSCLC). This review included an extensive search of electronic databases, such as PubMed (U.S. National Library of Medicine, Bethesda, MD, USA), Embase (Elsevier, Amsterdam, The Netherlands), and the Cochrane Library (The Cochrane Collaboration, London, UK). Key search terms included “non-small cell lung cancer”, “elderly patients”, “immune checkpoint inhibitors”, “efficacy”, and “safety”. Studies were included if they focused on ICIs in elderly patients with NSCLC and reported efficacy outcomes (overall survival, progression-free survival, and response rates) or safety outcomes (adverse events and quality of life), while those focused exclusively on younger populations or lacking age-specific data were excluded. Relevant studies were selected based on their relevance and contribution to understanding the efficacy and safety of ICIs in the target population. Data were extracted and synthesized qualitatively to provide a comprehensive overview of the current evidence. The review aimed to summarize and integrate findings on treatment outcomes and address gaps in the literature related to immunosenescence and its effects on treatment efficacy and safety. Data were synthesized qualitatively, with a focus on summarizing the efficacy and safety outcomes of ICIs in elderly patients with NSCLC. The review also included an assessment of immunosenescence and its potential impact on treatment outcomes.

## 3. Older Patients

The term “elderly” is used to describe people aged sixty and over [34]. the use of this term can perpetuate negative stereotypes and ageist discourses, leading to social exclusion and discrimination against older people [35]. Chronological age represents the number of years a person has lived, while physiological age reflects their functional status, including factors like organ function, mobility, and overall health. In oncology, understanding physiological age is crucial for tailoring cancer treatments to individual needs [36]. Physiological age considers divergences in rates of age-related decline in performance that can be measured using biological markers such as telomere length, DNA methylation, and gene expression patterns. Aging has been shown to vary significantly from person to person, and, in many cases, physiological changes may not necessarily match an individual’s chronological age [36]. One must understand physiological age not only in the healthcare field but also in decision making regarding cancer treatment. This knowledge helps to come up with tailor-made solutions that consider the individual’s level of functioning and potential risks [37]. Considering someone’s physiological age in examining their chronological age facilitates preventive measures, diagnostic screenings, and treatment decisions by personalizing them and making them effective. Such consideration guarantees that interventions and treatments are suitable for the unique individual needs and capacities [38,39].

## 4. The Particularity of the Elderly Subject

In elderly patients with lung cancer, specific clinical characteristics are observed, with a median age of diagnosis around 79 years. More than half of these patients are over the age of 80 [40]. A significant proportion are diagnosed with squamous cell carcinoma, the most common form of lung cancer in this age group, and many demonstrate good performance status (PS), which is often assessed using the Eastern Cooperative Oncology Group (ECOG) scale to evaluate a patient’s level of functioning and ability to carry out daily activities [41]. Despite this, the majority of older adults with lung cancer are often not provided with adequate care, and appropriate treatment options may be lacking for this subgroup [42]. There is a growing interest in studying immunosenescence and its relationship with the administration of ICIs in older patients with NSCLC. Treatment strategies for elderly lung cancer patients might need to be adapted compared to those used for younger patients [41,43,44,45,46]. However, older patients often have a delayed diagnosis of lung cancer [47]. One reason for this could be the under-diagnosis of pathology—a significant number of older patients receive only a clinical rather than a pathological diagnosis [33]. Furthermore, there are lung cancer-specific vulnerabilities that, for instance, are not related to age per se but to frailty in older adults, which may lead to delayed diagnosis and treatment [48]. Nevertheless, older patients receive less active treatment than their younger counterparts [49]. On the other hand, older patients in good physical condition can benefit optimally from these intensive treatment strategies as younger patients [50]. Elderly patients who have lung cancer might be far more vulnerable than healthy elderly. More research is needed to assess the vulnerability of older lung cancer populations thoroughly, and immediate action is needed to ascertain that the entire spectrum of treatment options is available.

## 5. Immunosenescence

The phenomenon of immunosenescence, or immune function decline that accompanies aging, affects lung cancer among elderly individuals [51]. Immunosenescence, characterized by age-related impairments in immune responses, profoundly impacts the development of immune reactions to both foreign and self-antigens. This phenomenon can be understood through three primary theories: autoimmunity, immunodeficiency, and immune dysregulation. Autoimmunity arises as aging leads to an accumulation of memory T cells and a decline in regulatory T cell function, resulting in increased autoantibodies and autoimmune diseases due to a loss of self-tolerance [52,53]. Immunodeficiency in the elderly is marked by reduced natural killer cell activity, a diminished pool of naïve T and B cells, and impaired antibody production against new antigens, which heightens susceptibility to infections [54,55,56]. Immune dysregulation is exacerbated by chronic low-grade inflammation and the senescence-associated secretory phenotype (SASP), leading to skewed immune responses and suboptimal vaccination outcomes [53,54]. In summary, immunosenescence might play a significant role in the impact of aging on cancer and immunity and influence response to ICIs among older adults. ICIs are an effective treatment for NSCLC, but their effectiveness varies depending on age, whereby older people may receive fewer benefits compared to younger ones [57]. Although they are a significant part of the patient population [58], there remains a lack of adequate inclusion of elderly patients in clinical trials. However, preliminary results suggest that ICIs-based immunotherapy can be effective and well tolerated among older individuals with advanced NSCLC [59,60]. The immune checkpoint inhibitors’ response to elderly patients depends upon various factors, such as the modified Glasgow prognostic score (mGPS) and comorbidities. The understanding of biological aspects of immunosenescence should be deepened, while reliable biomarkers for accurate prediction of immune reactions are sought. Biomarkers are critical in optimizing ICIs for elderly patients with lung cancer, particularly given the variability in treatment responses. The established biomarker, PD-L1, has limitations in predictive accuracy, necessitating the exploration of additional biomarkers to enhance patient selection and treatment efficacy [61,62]. Recent advancements in technologies such as liquid biopsies and multimodal analyses have facilitated the identification of novel circulating biomarkers and comprehensive biomarker signatures that may better predict responses to ICIs [63,64]. These emerging biomarkers, including tumor mutation burden and immune cell profiling, are essential for tailoring therapies, especially in the elderly population, which may experience different immune responses and treatment tolerances [65].

## 6. Efficacy of Immunotherapy in Older Patients

### 6.1. Results of Phase 3 Studies. Subgroup Analysis in Those > 65 Years Old

The KEYNOTE-010 trial evaluated pembrolizumab vs. docetaxel in patients with relapsed NSCLC expressing PD-L1 at levels of 1% or higher. Pembrolizumab significantly improved overall survival and progression-free survival compared to docetaxel, with higher objective response rates. The treatment with pembrolizumab was associated with a lower incidence of severe adverse events. The benefit was evident across different age groups, although the study did not include patients over 70 years old. Among those aged 65 and older, the treatment’s efficacy was somewhat lower compared to younger patients, though this could be influenced by other factors beyond age alone [66]. In the KEYNOTE-024 trial, first-line treatment with pembrolizumab was compared to platinum-based chemotherapy in patients with advanced NSCLC and high PD-L1 expression. Pembrolizumab showed substantial improvements in overall survival and progression-free survival compared to chemotherapy. The treatment also resulted in fewer severe adverse events. Although older patients experienced considerable benefits, the advantage was more pronounced in those younger than 65 years. The impact on quality of life favored pembrolizumab, with sustained improvements noted over time compared to chemotherapy [67,68]. The KEYNOTE-042 trial evaluated pembrolizumab monotherapy vs. standard chemotherapy in patients with advanced NSCLC expressing PD-L1. The study revealed a significant improvement in overall survival with pembrolizumab, particularly in patients with high PD-L1 expression levels. Importantly, pembrolizumab was associated with a lower incidence of severe adverse events compared to chemotherapy. Survival benefits were consistently observed across all age groups, including elderly patients, reinforcing its potential as a standard of care in this population [69]. In the phase III KEYNOTE-189 trial, patients with advanced NSCLC who were treatment-naïve were randomly assigned to receive a combination of platinum-based chemotherapy, pemetrexed, and either pembrolizumab or a placebo. The results indicated that the combination treatment including pembrolizumab significantly improved overall survival compared to the placebo. This benefit was observed across various age groups, though it was more pronounced in patients younger than 65 years compared to those over 65. Both groups experienced a similar rate of severe adverse events, but discontinuation rates due to these events were higher in the pembrolizumab group compared to the placebo group [70,71]. The IMpower150 trial examined the addition of atezolizumab to a regimen of bevacizumab and chemotherapy in advanced NSCLC. This combination therapy demonstrated superior overall survival compared to chemotherapy alone, with consistent efficacy observed across diverse patient subgroups, including older individuals. However, the regimen was associated with a slightly higher incidence of adverse events, emphasizing the necessity of evaluating patient tolerance, particularly in elderly populations, before initiating treatment [72]. The CheckMate-017 trial investigated nivolumab vs. docetaxel in patients with relapsed NSCLC. Nivolumab led to longer overall survival and progression-free survival, as well as a higher response rate compared to docetaxel. The safety profile of nivolumab was better, with fewer severe adverse events. The treatment benefited patients up to age 74, but the advantage was not as clear in those aged 75 and older, likely due to the small size of this subgroup. Quality of life improvements were significant with nivolumab, with better outcomes sustained compared to docetaxel [73]. Data from multiple clinical trials involving nivolumab in solid tumors, including CheckMate-057 and CheckMate-017, were combined in a recent study [74]. The safety data were analyzed based on three age groups: <65, ≥65, and ≥70 years. The study found that the incidence of any grade 3/4 adverse events (AEs) was 58.4%, 62.6%, and 71.7% in the respective age groups. However, the assessment of treatment-related AEs was not included in the analysis [75]. Also, the comparison of AE incidence rates is severely confounded by follow-up variations and competing risk with death when comparing age groups. In the CheckMate-026 trial, nivolumab was tested against platinum-based chemotherapy in patients with PD-L1-positive NSCLC. The results showed that chemotherapy provided a longer progression-free survival, and there was no significant difference in overall survival between nivolumab and chemotherapy. Nivolumab continued to have a more favorable safety profile with fewer severe adverse events. For older patients, nivolumab’s efficacy in terms of survival was comparable to that observed in younger patients. Real-world data for elderly patients indicated varying response rates and survival outcomes, though comparisons with younger patients were lacking [76,77]. In the CheckMate-227 trial, a dual immunotherapy regimen combining nivolumab and ipilimumab was compared with chemotherapy as a first-line treatment for advanced NSCLC. This approach provided durable survival benefits, surpassing those achieved with chemotherapy. Subgroup analyses further confirmed the efficacy of the combination in older patients. Nevertheless, the higher rate of treatment-related adverse events necessitates careful patient selection and a thorough assessment of fitness before initiating dual immunotherapy [78]. The OAK study compared atezolizumab to docetaxel in pre-treated patients with advanced NSCLC. Atezolizumab notably improved overall survival compared to docetaxel, with efficacy observed in both PD-L1-positive and PD-L1-negative patients. The safety profile was advantageous for atezolizumab, with a lower incidence of severe adverse events. The survival benefit of atezolizumab was seen across all age groups, including older patients, though detailed subgroup data for the older population were not available [74,77,79,80,81,82].

### 6.2. Results of Phase 3 Studies. Pooled Analysis in Those > 75 Years Old

A pooled analysis by Akinboro O and colleagues evaluated the outcomes of patients with advanced NSCLC who had high PD-L1 expression (≥50%) and received either chemotherapy combined with immune checkpoint inhibitors (chemo-ICI) or ICIs alone. The analysis suggested that, for most patient subgroups, the survival outcomes (overall survival and progression-free survival) with chemo-ICI regimens were comparable to or better than those achieved with ICI-only regimens. However, for patients aged 75 years and older, the addition of chemotherapy did not demonstrate improved outcomes compared to ICIs alone. These findings emphasize the importance of personalized treatment decisions that weigh the potential benefits and risks of incorporating chemotherapy into ICI regimens, considering individual patient factors that could affect treatment tolerance and efficacy. Pembrolizumab has significantly enhanced overall survival in elderly patients with advanced NSCLC with high PD-L1 expression. Specifically, for those with a PD-L1 tumor proportion score (TPS) of 1% or greater, and even more so for those with a TPS of 50% or higher, pembrolizumab offers a notable survival advantage compared to standard treatments. When used as a first-line therapy, pembrolizumab also demonstrates superior overall survival for elderly patients with high PD-L1 expression compared to chemotherapy alone. Additionally, pembrolizumab is associated with a lower frequency of treatment-related adverse events in elderly patients than chemotherapy, indicating a more favorable safety profile [83] (Table 1).

## 7. Current Decision-Making Tools

In managing patients with NSCLC who lack oncogenic drivers, two primary tools are used to guide treatment decisions: Performance status (PS) and the G8 geriatric assessment, a screening tool designed to identify frailty and predict outcomes in elderly cancer patients. These tools evaluate overall health and functional status, helping to determine appropriate treatment options [84]. They are employed to gauge the patient’s overall condition, which aids in selecting the most suitable therapies [85]. The Aura guidelines recommend the use of both PS and G8 for managing lung cancer [86]. Performance Status (PS) plays a crucial role in the management of lung cancer. Patients with a poor PS (ECOG PS ≥ 2) tend to have significantly worse survival rates and are more likely to require extensive healthcare services compared to those with a better PS (ECOG PS < 2). In cases of advanced non-small cell lung cancer (NSCLC), patients with a PS of 2 or higher who received pembrolizumab monotherapy showed lower rates of disease control, shorter progression-free survival (PFS), and reduced overall survival (OS) compared to those with a PS of less than 2 [87,88]. The G8 tool, specifically designed for elderly patients, assesses various aspects such as mobility, nutrition, and cognition [89] (Figure 1).

## 8. Discussion

Elderly patients pose a challenging population as they are likely to suffer from various complications, such as a higher risk for serious side effects due to their existing conditions, decreased physical function, organ failure, cognitive decline, and loss of social network support. The exclusion of the elderly from immunotherapy trials is a topic that needs further research because these trials have the potential to be of great use for this population. Additionally, given the growing number of cancer patients who are 70 years or older, there is likely to be a greater disparity between the trial population and the actual population being treated in everyday clinical practice. The latter group may experience higher rates of treatment discontinuation as well as poorer performance status (PS) [90]. The safety profile of immune checkpoint inhibitor (ICI) therapy remains uncertain concerning whether older patients are at greater risk for severe immune-related adverse events (irAEs) compared to younger patients undergoing single-agent immunotherapy. Some research suggests that elderly patients may experience increased pulmonary toxicity and a higher likelihood of irAE recurrence when rechallenged with ICIs [91]. Additionally, a comprehensive meta-analysis of various cancers treated with anti-CTLA-4 or anti-PD-(L)1 agents found that while rare, fatal toxicities were more prevalent among older patients than their younger counterparts [92]. It is crucial to recognize that while older adults may have an increased susceptibility to autoimmunity, this does not automatically lead to the onset of autoimmune diseases, and such susceptibility can exist independently of cancer status [93,94]. Most studies on immunotherapy in elderly patients have concentrated on immune checkpoint blockade (ICB) monotherapy. The introduction of ICBs in combination with chemotherapy as a first-line treatment underscores the need for further research and safety data tailored to different age groups. Although current clinical evidence is sparse, older patients may exhibit a less favorable response to chemo-immunotherapy and could face a higher risk of severe adverse events (AEs) and treatment discontinuation related to AEs when treated with ICBs and chemotherapy [95,96]. However, it remains unexplored whether the increased rate of treatment discontinuation correlates with poorer survival outcomes.

Fighting cancer among the elderly has been a focus of national strategies since 2009, with attention paid to the organization of care, research, and assistance to patients. The development of oncogeriatric coordination units has as its goal the improvement of old-age people’s care and the reduction of disparities between regions [97]. Nevertheless, there remain several obstacles that must be addressed in the care of older cancer patients. The lack of a sufficient number of geriatricians and clinical oncologists poses a significant barrier to comprehensive care [98]. Furthermore, there is a need for patient-centered care in managing cancer among older adults. This should include the application of frailty screening tools, geriatric assessments, and shared decision making [99]. It is important to promote health literacy and also involve patient-experts in their healthcare journey [100]. The involvement of dedicated partners, such as case-management nurses and specialized pharmacists, plays a vital role in tailoring the care pathway for older adults with cancer [101].

Because of the special demands and possibilities in the case of elderly people, studies on elderly individuals with lung cancer are essential. Elderly lung cancer patients are often discriminated against due to their age and thus have treatment withheld from them, despite evidence that shows that they too can benefit from surgical resection and adjuvant therapy [102]. Studies have been conducted to establish the safety and feasibility of open anatomical lung resections, which resulted in similar outcomes for survival in younger patients [103]. Immunotherapy is also efficient and safe among aged patients as it is seen in young ones, showing no major variation in objective response rate or progression-free survival [104]. Therefore, more investigations must be carried out to discover the predictors of immunotherapy for the elderly population as well as to collect valid data representing this population. Age alone is not enough to decide on treatment, but personalized approaches based on functional age and geriatric assessment scales should also be considered.

Considering the special issues and specific needs of older people with lung cancer, there should be a biomarker designed especially for this age group. Some studies have found that aging has a strong impact on the outcome and benefits of treatment in cases of lung adenocarcinoma [105]. However, seniors are often overlooked by clinical trials, which means a lack of information on whether ICIs are safe and effective in such patients [13]. Nevertheless, given that the majority of NSCLC cases occur in elderly individuals, having a successful predictive regimen for this cohort may greatly improve overall survival and progression-free survival rates [106]. Lung cancer development risk is to be gauged using the help of biomarkers, which can also be applied in screening, diagnosis, monitoring, and prognosis prediction. It may also help to customize a patient’s treatment plan and even take a closer look at some high-risk groups for early detection of lung cancer vs. benign pulmonary lesions [51]. Therefore, the biomarkers specifically designed for old people with lung cancer must be developed to tailor prognosis prediction and immunotherapeutic strategies along with personalized therapy plans [107].

## 9. Conclusions

The management of elderly patients with advanced NSCLC who do not have targetable molecular alterations should be personalized based on individual assessments. These assessments should consider factors such as expected survival prognosis, risk of toxicity, quality of life, and patient preferences. It is important to avoid unjustified under-treatment. Including elderly patients in oncology clinical trials is a critical issue that needs to be addressed. Therefore, future oncology studies should likely focus on including patients based on both age and “geriatric” or functional criteria rather than age alone. This approach assumes that “fit” elderly patients can be treated similarly to younger patients. Such patient profiling would also enhance the characterization of the study population, potentially allowing for the identification of specific health domains that, when impaired, might be linked to reduced benefits from experimental treatments.

## Figures and Tables

**Figure 1 ijms-26-02120-f001:**
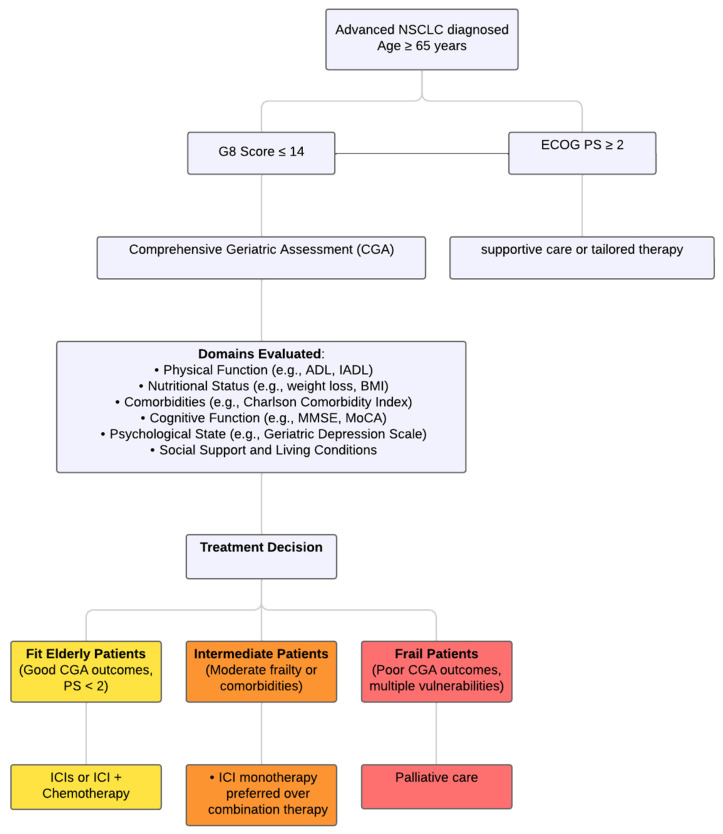
Flowchart of the Comprehensive Geriatric Assessment Process for Elderly Patients with NSCLC, Highlighting Key Decision Points and Treatment Pathways Based on Functional and Physiological Assessments.

**Table 1 ijms-26-02120-t001:** Summary of key clinical trials on ICIs in elderly patients with NSCLC.

Trial	Population	Intervention	Efficacy Outcomes	Safety Outcomes	Key Findings for Elderly Patients
KEYNOTE-010	Relapsed PD-L1-positive (≥1%) NSCLC (41% ≥65 years; no patients >70 years)	Pembrolizumab (2 or 10 mg/kg) vs. Docetaxel	Median OS: 10.4−12.7 vs. 8.5 months (HR 0.71, *p* = 0.0008). ORR: 18% vs. 9%. PFS benefit only for PD-L1 ≥50%.	Grade 3−5 AEs: 13–16% (Pembrolizumab) vs. 35% (Docetaxel). No age-specific safety data.	Similar survival benefit for older (HR 0.76) and younger (HR 0.63) patients, with lower toxicity for Pembrolizumab.
KEYNOTE-024	Advanced NSCLC, PD-L1 ≥50% (subgroup for older patients analyzed)	Pembrolizumab (200 mg) vs. Chemotherapy	Median OS: 30 vs. 14.2 months (HR 0.63, *p* = 0.002). Median PFS: 10.3 vs. 6.0 months (HR 0.5, *p* < 0.001).	Grade 3–5 AEs: 26.6% (Pembrolizumab) vs. 53.3% (Chemotherapy). Better QoL with Pembrolizumab (*p* = 0.002).	Stronger PFS benefit for older patients (HR 0.45) than younger (HR 0.61), but confounders may affect interpretation.
KEYNOTE-042	Advanced NSCLC, PD-L1 ≥1% (33% ≥65 years)	Pembrolizumab vs. Chemotherapy	Median OS: 16.7 vs. 12.1 months (HR 0.81, *p* = 0.0018). Subgroup with PD-L1 ≥50%: HR 0.53.	Lower incidence of grade ≥3 AEs with Pembrolizumab (17.8% vs. 41% for chemotherapy).	Survival benefit consistent across older (HR 0.77) and younger patients (HR 0.81).
KEYNOTE-189	Treatment-naïve advanced non-squamous NSCLC (median age: 65 years)	Platinum–pemetrexed + Pembrolizumab vs. Placebo	OS (1-year): 69.2% vs. 49.4% (HR 0.49, *p* < 0.001). Survival benefit across all age groups.	Grade 3–5 AEs: ~66% for both arms. Discontinuation rates: 13.8% (Pembrolizumab) vs. 7.9% (Placebo).	Less pronounced OS benefit in older patients (HR 0.64) vs. younger patients (HR 0.43). High AE rates in both groups.
IMpower150	Advanced NSCLC (median age: 63 years)	Atezolizumab + Bevacizumab + Chemotherapy vs. Bevacizumab + Chemotherapy	Median OS: 19.2 vs. 14.7 months (HR 0.78). Survival benefit across all age groups, including elderly.	Grade 3–4 AEs: 42% (Atezolizumab arm) vs. 36% (control). No age-specific data provided.	Consistent OS benefit across age groups, including those ≥65 years.
CheckMate-017	Relapsed squamous NSCLC (44% ≥65 years)	Nivolumab vs. Docetaxel	Median OS: 9.2 vs. 6.0 months (HR 0.59, *p* < 0.001). Median PFS: 3.5 vs. 2.8 months (HR 0.62, *p* < 0.001). ORR: 20% vs. 9%.	Grade 3–4 AEs: 7% (Nivolumab) vs. 55% (Docetaxel). Significant QoL improvement for Nivolumab.	OS benefit consistent for patients up to 74 years but absent in ≥75 subgroup (HR 1.85). Small subgroup limits conclusions.
CheckMate-026	PD-L1-positive (≥1%) NSCLC (48% ≥65 years)	Nivolumab vs. Platinum-based chemotherapy	Median OS: 14.4 vs. 13.2 months (HR 1.09, *p* = 0.25). Median PFS: 5.9 vs. 4.2 months.	Grade 3–4 AEs: 18% (Nivolumab) vs. 51% (Chemotherapy). Discontinuation rates lower for Nivolumab.	OS and PFS outcomes similar across age groups. Real-world data indicate modest ORR and median OS for patients ≥75 years.
CheckMate-227	Advanced NSCLC (43% ≥65 years)	Nivolumab + Ipilimumab vs. Chemotherapy	Median OS: 17.1 vs. 14.9 months (HR 0.73). Durable response observed in PD-L1 ≥1%.	Grade 3–4 AEs: 31% (Nivolumab + Ipilimumab) vs. 36% (Chemotherapy).	Consistent OS benefit in older (HR 0.78) and younger (HR 0.71) patients, though toxicity was higher in combination therapy.
OAK	Pre-treated advanced NSCLC (47% ≥65 years)	Atezolizumab vs. Docetaxel	Median OS: 13.8 vs. 9.6 months (HR 0.73, *p* = 0.0003). PD-L1-independent OS benefit.	Grade 3–5 AEs: 15% (Atezolizumab) vs. 43% (Docetaxel). Favorable safety profile for Atezolizumab.	OS benefit seen across all age groups, with greater benefit in older patients (HR 0.66 vs. HR 0.80 for younger patients).
Pooled Analysis	Advanced NSCLC, PD-L1 ≥50%	Chemo-ICI vs. ICI Monotherapy	OS/PFS comparable for ICI monotherapy vs. chemo-ICI in patients ≥75 years.	Chemo-ICI associated with higher toxicity in older patients.	Supports ICI monotherapy for patients ≥75 years due to reduced toxicity without loss of efficacy.

## Data Availability

Not applicable.

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
