# Peer review of "Immunotherapy for Elderly Patients with Advanced Non-Small Cell Lung Cancer: Challenges and Perspectives"

_ijms, 2025, doi:10.3390/ijms26052120_

Round 1

Reviewer 1 Report

Comments and Suggestions for Authors

The manuscript focused on Immunotherapy for Elderly Patients with Advanced NSCLC: Challenges and Perspectives represents a timely relevant and technically correct manuscript requiring minor comments to be accepted for the publication on this journal

- In the introduction section, please, could the authors point on the clinical advantages of ICIs in LC patients?

- In the manuscript, please, could the authors consider how integrated strategies (ICIs plus radio chemo) may be helpful for these patients

- In the text, please, could the authors consider promising biomarkers able to stratify elder patients under ICIs regimen?

Author Response

We sincerely appreciate your insightful recommendations and valuable suggestions. After carefully considering all of your feedback, we have made significant revisions to the relevant sections in this updated version. We hope these changes effectively address your concerns and enhance the overall quality of the manuscript.

Comment 1: In the introduction section, please, could the authors point on the clinical advantages of ICIs in LC patients?

Response 1: Thank you for your observation. We added a paragraph in the introduction to explicitly discuss the clinical advantages of immune checkpoint inhibitors (ICIs), emphasizing their ability to improve overall and progression-free survival with a favorable safety profile compared to traditional chemotherapy, particularly in elderly patients with advanced NSCLC. Please refer to page 2, lines 53-58.

Comment 2: In the manuscript, please, could the authors consider how integrated strategies (ICIs plus radio chemo) may be helpful for these patients.

Response 2: Thank you for your insightful comment. The PACIFIC trial demonstrated the efficacy of concurrent chemoradiotherapy followed by durvalumab in patients with stage III NSCLC. In contrast, the focus of this review is on advanced (stage IV) NSCLC, where systemic therapies, particularly immune checkpoint inhibitors (ICIs), play a central role in treatment."

Comment 3: In the text, please, could the authors consider promising biomarkers able to stratify elder patients under ICIs regimen?

Response 3: Thank you for highlighting this point. A subsection was added to the discussion addressing biomarkers like PD-L1 expression, tumor mutational burden (TMB). Their relevance for stratifying elderly patients and predicting response to ICIs was elaborated upon. Please refer to page 5, lines 204-213.

Reviewer 2 Report

Comments and Suggestions for Authors

This is an excellent succinct review on the efficacy of immune checkpoint blockade immunotherapy in aged NSCLC patients. A few minor points:

On page 2, (i) there comes suddenly a sentence in French. It should be removed, (ii) what is CELLDRIFT signature, define it, (iii) SASP is not a substance, it is a cell phenotype with certain features.

On page 4/5, define performance status, geriatric assessment tool and ECOG.

Page 5, what does PDL-1 percentage of expression mean?

Figure 1. All abbreviations used in the Figure should be explained.

Author Response

We sincerely appreciate your thoughtful feedback and constructive suggestions. After thoroughly reviewing your comments, we have made substantial revisions to the relevant sections in this updated version. We believe these changes effectively address your concerns and improve the overall clarity and quality of the manuscript.

Comment 1: On page 2, (i) there comes suddenly a sentence in French. It should be removed, (ii) what is CELLDRIFT signature, define it, (iii) SASP is not a substance, it is a cell phenotype with certain features.

Response 1:

  • The French sentence on page 2 has been removed. Please refer to page 2, line 86.
  • A brief explanation of the CELLDRIFT signature, highlighting its role in distinguishing cancerous from normal tissues in aging-related cancers, has been included. Please refer to page 2, lines 87-91.
  • The text was revised to describe SASP as a phenotype characterized by the secretion of pro-inflammatory cytokines, chemokines, and proteases that contribute to tumor progression. Please refer to page 2, lines 93-97.

Comment 2: On page 4/5, define performance status, geriatric assessment tool and ECOG.

Response 2: For clarity, the definitions for "performance status," "geriatric assessment tool," and "ECOG" have been added. Please refer to pages 4, 160-162, and 8, 311-313.

Comment 3: Page 5, what does PDL-1 percentage of expression mean?

Response 3: We clarified that PD-L1 percentage of expression refers to the proportion of tumor cells expressing PD-L1, which is used as a predictive biomarker for response to ICIs. Please refer to Table 1 and page 2, lines 70-74.

Comment 4: Figure 1. All abbreviations used in the Figure should be explained.

Response 4: all abbreviations in Figure 1 have been added for better understanding. Please refer to the abbreviations on page 11, line 422.